# EEGTRANS: TRANSFORMER-DRIVEN GENERATIVE MODELS FOR EEG SYNTHESIS

## ABSTRACT

Recent advancements in Large Language Models (LLMs) have been significant, largely due to improvements in network architecture, particularly the transformer model. With access to large training datasets, LLMs can train in an unsupervised manner and still achieve impressive results in generating coherent output. This study introduces a transformer-based generative model, EEGTrans, designed for sequentially generating synthetic electroencephalogram (EEG) signals. Given the inherent noise in EEG data, we employ a quantized autoencoder that compresses these signals into discrete codes, effectively capturing their temporal features and enabling generalization across diverse datasets. The encoder of EEGTrans processes EEG signals as input, while its decoder autoregressively generates discrete codes. We evaluate our method in a motor imagery Brain-Computer Interface (BCI) application, where merging data across datasets is particularly challenging due to experimental differences. Our results demonstrate that the synthetic EEG data effectively captures temporal patterns while maintaining the complexity and power spectrum of the original signals. Moreover, classification results show that incorporating synthetic data improves performance and even surpasses that of models based on Generative Adversarial Networks (GANs). These findings highlight the potential of transformer-based generative models to generalize effectively across multiple datasets and produce high-quality synthetic EEG signals. The source code is available at https://anonymous.4open.science/r/EEGTrans-Transformer-Driven-Generative-Models-for-EEG-Synthesis-0FD9/.

## 1 INTRODUCTION

Large language models (LLMs) have been extensively utilized across various scenarios due to their powerful model characteristic: the generative models. These models are not restricted to producing specific forms of output; instead, they can generate output in any form. The progress of LLMs is largely attributed to the implementation of the attention mechanism (Vaswani et al., 2017). This mechanism enables the processing of long-range dependency inputs and can be adapted to numerous domains, such as text-to-image generation (Ramesh et al., 2021) and speech-to-text generation (Radford et al., 2023). Recent studies demonstrate that transformer models can generate activations resembling those observed in the human brain (Schrimpf et al., 2021; Caucheteux & King, 2022). Additionally, models that perform well in predicting the next word in a sequence also show proficiency in predicting brain measurements. This computational evidence highlights the crucial role of predictive processing in shaping the brain's comprehension of language. This leads to a new question: Can transformer architecture effectively generate brain signals?

The most common generative models used in Electroencephalography (EEG) research are Generative Adversarial Networks (GANs) (Goodfellow et al., 2014). Researchers have extensively explored the application of GANs to various EEG fields, including motor imagery (Hartmann et al., 2018; Xu et al., 2021; Fahimi et al., 2020), emotion recognition (Luo & Lu, 2018; Luo et al., 2020), epileptic seizure detection (Wei et al., 2019; Rasheed et al., 2021), etc. Many of these applications aimed to address the data scarcity problem by generating synthetic EEG data, often incorporating both Conditional Generative Adversarial Networks (CGAN) (Mirza & Osindero, 2014) and Wasserstein Generative Adversarial Networks (WGAN) (Arjovsky et al., 2017). However, GAN models encounter a limitation when applied to EEG, as they lack inherent temporal generation processes (Bird et al., 2021). The generated output adheres to a fixed format and cannot extend indefinitely

to accommodate varying sequence lengths. This limitation might pose challenges for EEG applications where the duration of signal generation is ambiguous. Moreover, in the generation process, the signal output in each timestep lacks influence on the subsequent one, whereas EEG possesses a high temporal resolution characteristic.

Brain-computer interfaces (BCIs) enable direct communication between humans and machines. EEG is a common method in BCI due to its mobility and millisecond-range temporal resolution. With the decreasing data collection costs over recent decades, the effort to collect more data has become more accessible (Wolpaw et al., 2002). While more public datasets are being made available, recent research has primarily focused on advancing sophisticated models to enhance BCI performance. The emergence of the transformer architecture (Vaswani et al., 2017), successfully applied in both computer vision (CV) and natural language processing (NLP) with models like the Vision Transformer (ViT) (Dosovitskiy et al., 2020) and Generative Pre-trained Transformer (GPT) (Brown et al., 2020), has led to its adoption in EEG as well (Song et al., 2022).

The scalability of these transformers is widely recognized, as they tend to perform better with larger datasets (Kaplan et al., 2020; Henighan et al., 2020; Zhai et al., 2022). However, unlike in CV and NLP, where data from various sources can be combined to create a larger dataset, this approach is not feasible in EEG research. This is mainly due to significant variations among data from different subjects within the same dataset (Morioka et al., 2015; Jayaram et al., 2016), as human brains differ from each other (Gu & Kanai, 2014), resulting in differences in recorded brain activity. Moreover, the inter-variation between datasets is much greater than the intra-variation within datasets. Each BCI field employs distinct experimental designs, and even within the same field, each dataset has unique experimental settings. Challenges arise when attempting to merge datasets due to differences in experimental tasks, setups, and even the equipment used. The key strength of the transformer architecture lies in its attention mechanism, which can effectively learn representations across various domains. An interesting idea arises from this concept: Can a generative model be developed that can be applied across multiple EEG datasets exploiting the transformer's capabilities?

In this paper, we introduce EEGTrans, a novel framework designed to train generative models using data from multiple sources, capable of generating high-quality synthetic EEG data from unseen datasets. When we refer to "unseen", we are exploring the potential of applying knowledge acquired from a previously seen dataset (source dataset) to generate synthetic data for a new dataset (target dataset). Our EEGTrans employs a quantized autoencoder to compress EEG data into discrete codes. This process imposes a more stringent penalty on the generative models, discouraging the learning of trivial solutions. We conducted experiments using two publicly available motor imagery datasets. The quality of synthetic data can be evaluated through visualization and measuring the differences between real and synthetic data. At the same time, the model's performance will be assessed based on the downstream classification task. The contributions of this work can be summarized as:

1. This paper introduces a novel framework that employs a transformer-based generative model trained on multiple datasets in motor imagery EEG. To the best of our knowledge, this is the first research effort to take such an approach.

2. We demonstrate that, without explicit training on the new dataset, EEGTrans can generate synthetic signals that closely resemble real data, suggesting that transformers can capture the EEG characteristics present in motor imagery datasets.

3. We introduce a new loss design that utilizes the synthetic data generated by EEGTrans to enhance downstream BCI classification performance.

## 2 RELATED WORKS

### 2.1 GENERATIVE MODELS FOR BIOLOGICAL SIGNALS

**GAN**   Numerous studies have explored the utilization of GANs for generating synthetic EEG data related to motor imagery. Hartmann et al. (2018) explored the feasibility of using Convolutional Neural Network (CNN) to train a GAN progressively for generating synthetic EEG data. By modifying the improved WGAN training, they could train a GAN in a stable manner to generate synthetic signals closely resembling real EEG signals from a single channel and a single subject, both in the time and frequency domains. Roy et al. (2020) utilized Bidirectional Long Short-Term Memory

(Bi-LSTM) to build GANs for generating synthetic EEG data from a single channel. However, the methods mentioned above did not provide information on the classification performance after generating synthetic data.

Given the limited amount of data available from stroke patients compared to that from normal patients, Xu et al. (2021) utilized Cycle-consistent Adversarial Networks (CycleGAN) (Zhu et al., 2017) to generate motor-imagery EEG data of stroke patients from normal patients, thereby enhancing the classification performance. Xie et al. (2021) combined Long Short-Term Memory Generative Adversarial Networks (LGANs), Multi-output Convolutional Neural Network (MoCNN), and an attention network to enhance motor imagery classification performance. Classifiers and GANs for EEG signals need subject-specific training due to inter-subject variation, though the referenced studies above did not explicitly mention this. Fahimi et al. (2020) proposed a novel approach based on the Conditional Deep Convolutional Generative Adversarial Networks (DCGANs) to generate subject-specific artificial EEG by training on subject-independent data. This is accomplished by appending a subject-specific feature vector to both the generator and discriminator during the training process.

**Transformer** GPT-2 models were utilized to produce synthetic biological signals (electromyography and EEG) (Bird et al., 2021). However, the process of preparing these biological signals for interpretation by GPT was not elaborated upon. They utilized pre-trained weights on GPT, yet GPT is pre-trained on NLP, where inputs consist of tokens, which is notably different from processing continuous EEG signals. Moreover, for $n$ classes of data, $n$ GPT-2 models are trained, making the process time-consuming, especially when integrating new classes or datasets, as it requires additional time. Niu et al. (2021) built upon the previous research by utilizing GPT-2 models to generate EEG signals, aiming to improve the prediction of epileptic seizures.

A similar work that uses transformers to learn generic representations across multiple EEG datasets is LaBraM (Jiang et al., 2024). However, LaBraM primarily focuses on learning representations and improving downstream classification tasks. In contrast, our study mainly investigates the generative properties of the transformer architecture, exploring whether motor imagery datasets share underlying features, and whether transformers have the capability to generate synthetic EEG signals that capture these features.

Currently, there are no methods that attempt to train a generative model using multiple source EEG datasets.

## 2.2 VECTOR QUANTIZATION

Vector quantization serves as a method to compress inputs into discrete codes while simultaneously ensuring the essential fidelity of the data is preserved (Gray, 1984). Recently, vector quantization has gained widespread usage in deep learning following the introduction of Vector Quantized Variational AutoEncoder (VQ-VAE) (Van Den Oord et al., 2017). For example, SoundStream (Zeghidour et al., 2021) and EnCodec (Défossez et al., 2022) both use a residual vector quantizer (RVQ) (Juang & Gray, 1982) to quantize the output of the encoder. Following previous works, DeWave (Duan et al., 2023) utilized VQ-VAE to encode EEG signals into discrete codes. They suggested that encoding EEG signals according to their proximity to the nearest neighbor in the codex book could decrease variations among different subjects, thus improving generalization across subjects. Drawing inspiration from DeWave, we integrate the residual vector quantizer into our research and design a novel architecture to leverage its capabilities, aiming to attain generalization across datasets.

## 3 METHOD

### 3.1 TASK DEFINITION

When working with multiple distinct datasets, we classify them into two categories: source and target datasets. Generative models are first trained on the source datasets to learn EEG characteristics, including the temporal and frequency information of various motor imagery classes. We then adapt these pre-trained models to generate synthetic EEG data for the target datasets.

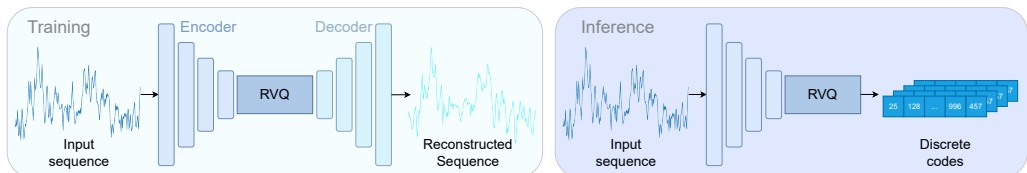

Figure 1: The architecture of the RVQ autoencoder model involves several stages (Section 3.2). In training, the autoencoder receives a single-channel EEG sequence and produces a reconstructed sequence, utilizing reconstruction loss to train the encoder, RVQ, and decoder. Later, the pre-trained encoder and RVQ are employed in generating discrete codes during inference.

## 3.2 RVQ

With EEG sequences $X \in \mathcal{R}^{C \times T}$, where $C$ represents the channels and $T$ signifies the timestamps, we utilize an RVQ to transform the continuous signals into discrete codes. An autoencoder is constructed for this process, comprising three modules: encoder, RVQ, and decoder, illustrated in Figure 1. Since the number of channels can differ among datasets, this procedure handles channels individually, one at a time. Initially, the encoder transforms a single-channel EEG sequence $x \in \mathcal{R}^T$ into embeddings $z_e(x)$. Subsequently, these embeddings are substituted with the latent variables $e$ corresponding to one of the codebooks in the RVQ. This is accomplished by calculating the distance between the embeddings and the latent variables and then substituting them with the latent variables that have the closest distance. Finally, the quantized embeddings are fed through the decoder to obtain the reconstructed sequence $x_r$. The autoencoder is trained using the reconstruction loss to minimize the distance between the input sequence and its reconstructed sequence, as well as to align their mean and standard deviation. Additionally, the codebooks are trained by minimizing the distance between the $\ell_2$-normalized latent variables and the embeddings as shown in Equation 1:

$$
\begin{aligned}
L_R &= ||x - x_r||_2^2 + ||\mu(x) - \mu(x_r)||_2^2 + ||\sigma(x) - \sigma(x_r)||_2^2, \\
L_{VQ} &= ||\text{sg}[z_e(x)] - e||_2^2 + \beta||z_e(x) - \text{sg}[e]||_2^2,
\end{aligned}
\tag{1}
$$

where the sg denotes the stop gradients. The encoder and RVQ are subsequently utilized to produce discrete code sequences for each EEG sequence, which will be employed in training the generative models.

## 3.3 EEGTRANS

We adopted the original dense transformer architecture (Vaswani et al., 2017) as our transformer-based generative model. Using this encoder-decoder architecture, we train an autoregressive model that takes EEG sequences as input and generates corresponding discrete codes. A similar setup can be found in Whisper (Radford et al., 2023), which was originally designed for speech-to-text translation. The input (continuous) and output (discrete) formats in Whisper align with ours. However, while spectrograms are commonly used as input in speech processing, this approach is less common in the EEG domain. For example, models like EEG Conformer (Song et al., 2022) and LaBram (Jiang et al., 2024) directly employ EEG signals. Therefore, we followed this approach and excluded spectrogram components from EEGTrans.

The architecture of the proposed EEGTrans utilized in this study is depicted in Figure 2. During the training process, EEGTrans is trained using the next code prediction task. Here, the decoder is tasked with predicting the code $Y_{t+1}$ corresponding to the next timestamp, based on the EEG inputs $x$ and the codes $Y_{\leq t}$ received up to the current timestamp. Once discrete codes are generated, we employ the decoder from the pre-trained RVQ autoencoder to get the signals $\hat{x}$ back. To obtain synthetic data that more closely represents the original data, we further fine-tune the pre-trained RVQ decoder. This fine-tuning occurs during EEGTrans training, where we continue to train the decoder alongside EEGTrans, all while keeping the encoder and codebooks frozen. The training

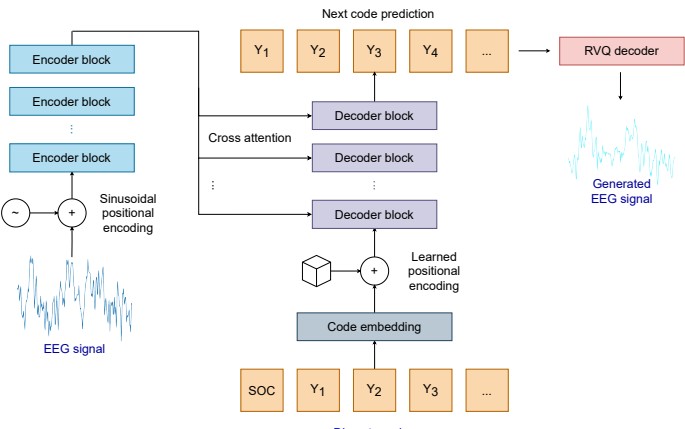

Figure 2: EEGTrans model architecture. EEGTrans takes EEG signals as inputs and generates discrete codes, the outputs of RVQ, in an autoregressive manner. This model is trained using the next code prediction task during the training phase. Once all the discrete codes are generated, the RVQ decoder is used to reconstruct the synthetic data.

Table 1: Datasets overview

| Dataset | No. of participants | Sampling rate (Hz) | No. of channels | Duration of each trial (s) | No. of classes |
|---|---|---|---|---|---|
| BCI Competition II Dataset III | 1 | 128 | 3 | 9 | 2 |
| BCI Competition IV Dataset 2b | 9 | 250 | 3 | 7 | 2 |
| BCI Competition IV Dataset 1 | 7 | 1000 | 59 | 6 | 2 |
| BCI Competition IV Dataset 2a | 9 | 250 | 22 | 6 | 4 |
| High Gamma Dataset | 14 | 500 | 128 | 4 | 3 |

loss for both models is defined in Equation 2:

$$L_{EEGTrans} = -\sum_{t=0}^{\tau-1} \log(p(Y_{t+1}|x, Y_{\leq t})),$$

$$L_{RVQ\ decoder} = ||x - \hat{x}||_2^2,$$

(2)

where $\tau$ represents timestamps within discrete token space, which may vary from timestamps $T$ based on the RVQ encoder design. Additionally, $t_0$ is a unique token $SOC$ added to denote the start of the code. We compared our proposed method to CycleGAN, a generative model that translates an input from a source domain to a target domain. The model architecture details can be found in the Appendix A.

## 4 EXPERIMENTS

### 4.1 DATASET

Three source datasets are utilized to train both RVQ autoencoder and the generative models, while two additional target datasets used to evaluate the models' performance and synthetic data quality. The datasets are outlined as follows: the source datasets include BCI Competition II Dataset III (Blankertz et al., 2004), BCI Competition IV Dataset 2b (Tangermann et al., 2012), and BCI Competition IV Dataset 1 (Tangermann et al., 2012); whereas the target datasets encompass BCI Competition IV Dataset 2a (Tangermann et al., 2012) and the High Gamma Dataset (Schirrmeister et al., 2017). Table 1 provides an overview of these datasets. Please refer to Appendix B.1 for a more detailed description.

## 4.2 DATA PREPROCESSING

As multiple datasets are utilized, it is essential to standardize them into a common format to facilitate model interpretation. For instance, various datasets may have different sampling rates, meaning that a fixed number of timestamps may represent varying durations across datasets. Consequently, preprocessing the data is crucial to enable model training across datasets. The first step is epoching: segmenting the complete EEG sequence of each dataset into epochs using event markers, preserving only the data occurring from the onset of the event to 2 seconds after the event onset. Subsequently, the data is resampled to 128 Hz, the lowest sampling rate among the five datasets used, using a fast Fourier transform. As the motor imagery field is chosen for validating the proposed method, only channels relevant to motor imagery are selected (refer to Appendix B.2 for more details). Finally, the signals within each epoch are normalized to zero mean and one standard deviation along the timestamp dimension. These processed data are then ready for training the generative models.

## 4.3 IMPLEMENTATION DETAILS

The RVQ autoencoder is built exclusively with a 1D convolutional layer (Conv1D) for the encoder, while both Conv1D and transpose Conv1D are employed for the decoder. The RVQ autoencoder is trained using the AdamW optimizer, with a learning rate of 1e-3 and a weight decay of 1e-4 for 1000 epochs. Please refer to Appendix C for further information on the configuration of the RVQ autoencoder.

The EEGTrans model architecture comprises an encoder and a decoder. The input embedding layer of the EEGTrans model consists of 6 Conv2D layers. The encoder block consists of 4 layers with an embedding size of 256 and 4 attention heads. The decoder block mirrors the settings of the encoder block, except for the input embedding layer, which is a simple lookup table storing embeddings of a fixed dictionary size. The output then passes through a multilayer perceptron (MLP) that maps the embeddings to discrete tokens. EEGTrans is trained using the AdamW optimizer with a cosine learning rate scheduler and a weight decay of 1e-3. The initial learning rate is set at 1e-6, with a warmup epoch of 20 and a maximum learning rate of 1e-3. Following 1000 epochs, the learning rate gradually decays to 1e-5. The RVQ decoder is trained with the AdamW optimizer with a constant learning rate 1e-3 and a weight decay of 1e-3.

The training of all models takes place on a single RTX 4090 GPU. Training for each epoch occurs sequentially, following the order: BCI Competition II Dataset III, BCI Competition IV Dataset 2b, and BCI Competition IV Dataset 1. Please refer to Appendix C for a more detailed description of the configurations for cycleGAN.

## 4.4 EVALUATION METRICS

We evaluate performance by visually inspecting the synthetic data and measuring differences between the real and synthetic datasets, including variations in the frequency domain and sample entropy. Furthermore, we utilize the synthetic data to train a classifier and assess whether it provides any advantages for downstream classification tasks. For EEGTrans, synthetic data is created by feeding real data into the encoder, which then prompts the decoder to iteratively generate the next timestamp code starting from the $SOC$ token. Then, the fine-tuned RVQ decoder is utilized to convert the discrete codes back into signals. CycleGAN employs generator $G$ to convert EEG signals directly into discrete codes, followed by the use of the RVQ decoder to reconstruct the signal.

## 4.5 DATA VISUALIZATION

We utilize EEGTrans to generate synthetic data for the BCI Competition IV Dataset 2a. In Figure 3, we provide a detailed comparison of the synthetic data produced by EEGTrans, presenting the real data alongside the corresponding synthetic data for Subject 1. We focus on three channels commonly used in motor imagery experiments, and two epochs are shown to allow us to confirm the robust performance of EEGTrans across different channels and multiple epochs. Additionally, we present the evoked data (averaged epoch data), further demonstrating EEGTrans's effectiveness for Subject 1. The visual comparison reveals a remarkable similarity between the synthetic and real data, highlighting EEGTrans's ability to generate high-quality synthetic data that closely matches the real data, with negligible visual differences in both the time and frequency domains.

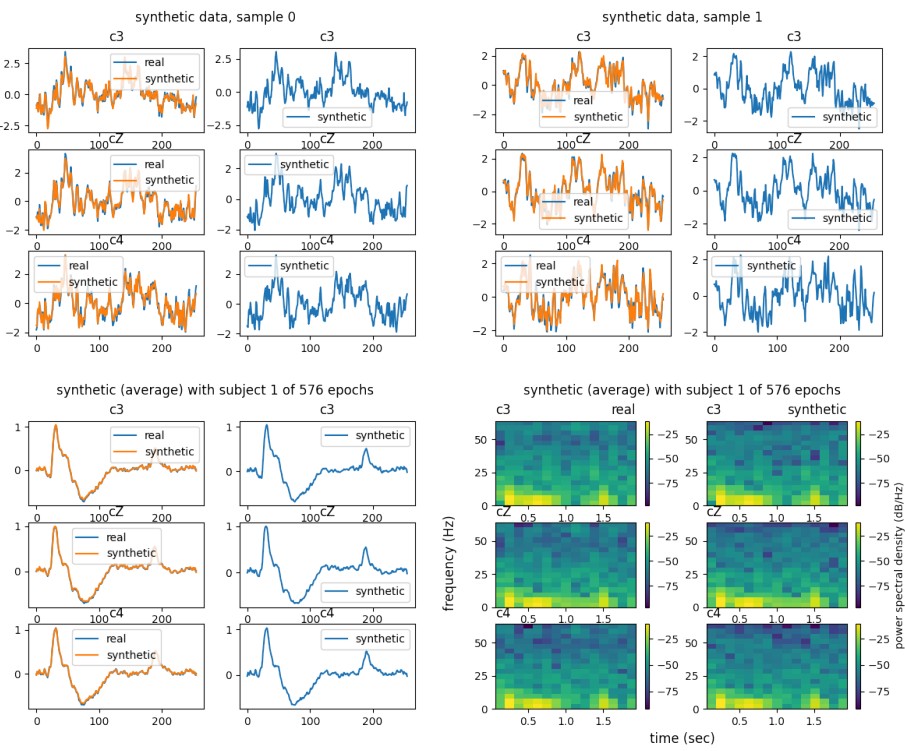

Figure 3: Data visualization. We visually inspect synthetic data generated by EEGTrans for Subject 1 in BCI Competition IV Dataset 2a. Both the real and synthetic data for the first two samples are displayed, revealing minimal differences between them. Furthermore, we illustrate the average of all epochs from Subject 1 in both the time and frequency domains.

Table 2: Spectral entropy and sample entropy comparison. Spectral entropy and sample entropy are used to assess the characteristics of a time series. The method that produces a value closest to the ground truth is considered the best in this case.

| Entropy | Method | Subject | | | | | | | | |
|---|---|---|---|---|---|---|---|---|---|---|
| | | 1 | 2 | 3 | 4 | 5 | 6 | 7 | 8 | 9 |
| Spectral | Ground truth | 4.70 | 5.33 | 5.46 | 4.79 | 4.61 | 4.65 | 4.61 | 5.29 | 4.45 |
| | EEGTrans | **4.61** | **5.16** | **5.25** | **4.70** | **4.67** | **4.59** | **4.56** | **5.16** | **4.46** |
| | CycleGAN | 4.08 | 4.44 | 4.46 | 4.20 | 4.10 | 4.21 | 4.16 | 4.55 | 4.36 |
| Sample | Ground truth | 1.55 | 1.79 | 1.89 | 1.54 | 1.67 | 1.42 | 1.42 | 1.76 | 1.23 |
| | EEGTrans | **1.50** | **1.69** | **1.78** | **1.48** | **1.59** | **1.39** | **1.39** | **1.68** | **1.27** |
| | CycleGAN | 0.96 | 1.06 | 1.08 | 0.98 | 0.94 | 0.95 | 0.95 | 1.09 | 0.97 |

However, it should be noted that high-frequency signals are not fully retained during synthetic data generation. Nevertheless, it is important to highlight that the most common frequency bands utilized in motor imagery decoding, namely the alpha (8-13 Hz) and beta (14-30 Hz) bands, are generally well-preserved. We also visualize the synthetic data generated by CycleGAN in Figure 4. While CycleGAN can produce synthetic data that exhibit trends somewhat similar to the real data, there are significant differences in magnitude. Additionally, the frequency distribution diverges from the original, displaying lower power across nearly all frequency bands. Visual inspection indicates that EEGTrans produces higher-quality synthetic data compared to CycleGAN. It is evident from the visual comparison that EEGTrans's generated data is significantly superior.

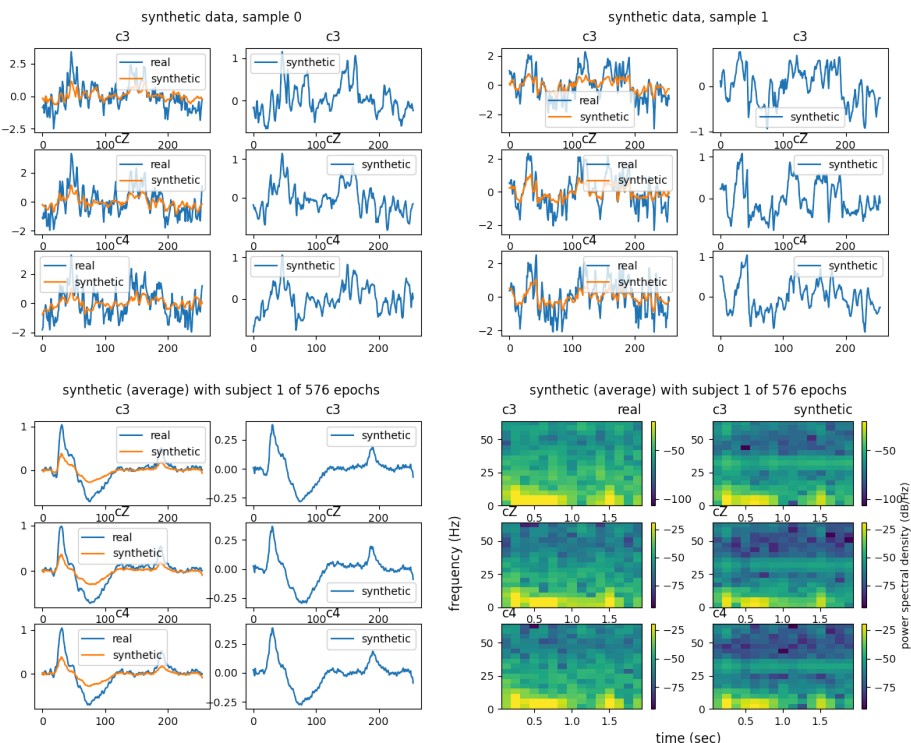

Figure 4: Visualization of synthetic data generated by CycleGAN for Subject 1 in the BCI Competition IV Dataset 2a.

### 4.6 TIME SERIES COMPLEXITY ANALYSIS

Besides visual inspection, we also calculate spectral entropy and sample entropy to verify synthetic data quality. Spectral entropy measures signal complexity or randomness in the frequency domain, derived from Shannon entropy applied to the power spectral density. Lower spectral entropy indicates power concentration at specific frequencies. Sample entropy quantifies the complexity and irregularity of time-series data, assessing the likelihood that similar patterns persist over time. Low sample entropy suggests the time series is more regular and predictable.

Spectral entropy and sample entropy are calculated for each time series. We report the values for each subject by averaging across all samples and all channels. As shown in Table 2, only EEGTrans closely matches the ground truth (real data) with minor differences, retaining high sample entropy and thus indicating high complexity and variation. However, it does not retain high-frequency components, which is evident in the spectral entropy. The synthetic data from CycleGAN loses power in important frequency bands for motor imagery and shows huge amplitude differences from the ground truth in the time domain. This might explain its low sample entropy, suggesting the synthetic data is not sufficiently representative.

### 4.7 BCI CLASSIFICATION TASK

To further validate the data quality, we utilize EEGNet (Lawhern et al., 2018), a widely used classification model in the EEG domain. This model has demonstrated its effectiveness in conducting classification tasks across various EEG applications and has emerged as a standard benchmark for comparison. In short, EEGNet is trained to perform a multi-class classification task separately on each target dataset. For detailed information about EEGNet, please refer to Appendix D.

In BCI Competition IV Dataset 2a, there are four classes for classification: left hand, right hand, both feet, and tongue. While generative models are trained without explicit labels, it is crucial to recognize that certain classes, like "tongue," may not have been present in the source datasets during

Table 3: Classification performance on BCI competition IV Dataset 2a (Section 4.7). R: real data; S: synthetic data; RS: combination of real and synthetic data; Aux: combination of real and synthetic data with auxiliary loss; *: p < 0.05.

| | | EEGTrans (%) | | | CycleGAN (%) | | |
|---|---|---|---|---|---|---|---|
| Subject | R | S | RS | Aux * | S | RS | Aux |
| 1 | 84.72 ±2.92 | 85.59 ±3.64 | **87.33** **±4.74** | 86.98 ±4.62 | 51.57 ±5.23 | 77.77 ±2.12 | 85.94 ±3.46 |
| 2 | 73.77 ±5.02 | 64.59 ±3.13 | 72.38 ±3.24 | **73.78** **±2.95** | 52.07 ±4.87 | 64.92 ±2.42 | 71.53 ±3.13 |
| 3 | 91.67 ±3.84 | 87.14 ±4.32 | 89.75 ±3.08 | **93.23** **±2.75** | 62.49 ±5.09 | 90.44 ±0.79 | 92.70 ±2.66 |
| 4 | 80.20 ±2.01 | 74.82 ±3.35 | 79.51 ±4.14 | **82.46** **±3.42** | 59.38 ±4.65 | 77.26 ±3.94 | 80.38 ±4.44 |
| 5 | 79.86 ±4.40 | 81.77 ±2.71 | **84.02** **±2.80** | 82.46 ±2.95 | 73.61 ±4.03 | 80.55 ±2.47 | 82.46 ±3.20 |
| 6 | 74.12 ±4.96 | 74.99 ±4.38 | **77.42** **±1.13** | 76.90 ±2.76 | 61.80 ±3.39 | 69.78 ±2.21 | 75.34 ±2.12 |
| 7 | **88.03** **±5.62** | 83.86 ±3.27 | 85.95 ±4.84 | 87.32 ±5.31 | 68.58 ±3.19 | 80.90 ±2.70 | 82.47 ±3.17 |
| 8 | 85.07 ±2.50 | 81.76 ±1.85 | **86.28** **±1.63** | 84.02 ±2.60 | 51.92 ±5.47 | 82.13 ±3.98 | 85.59 ±2.34 |
| 9 | 90.27 ±2.08 | 91.67 ±1.14 | 90.79 ±2.03 | **92.54** **±2.52** | 65.79 ±4.63 | 91.31 ±1.35 | 90.45 ±3.70 |
| Mean | 83.08 ±6.17 | 80.69 ±7.64 | 83.71 ±5.74 | **84.41** **±6.11** | 60.80 ±7.42 | 79.45 ±8.06 | 82.98 ±6.34 |

training. Table 3 shows the classification accuracy achieved through various approaches: using only real data, only synthetic data, combining real and synthetic data, and incorporating real data with synthetic data along with auxiliary loss. For a comprehensive analysis, we employ five-fold cross-validation instead of the original train-test split used in the competition. We then report each subject's mean and standard deviation of classification accuracy. We conduct comparisons between EEGTrans and CycleGAN across all these scenarios.

We designate "using only real data" as the benchmark. In the case of "using only synthetic data," synthetic data corresponding to the training index in that fold is utilized as training data, aligning with the benchmark. This approach ensures that no synthetic data corresponding to the testing index is used for training. For the "combining real and synthetic data" case, both real data and synthetic data of the training index are used in training, effectively doubling the number of training data compared to the benchmark. Lastly, it is important to note that instead of simply adding synthetic data as training data, we introduce a regularization term or sample weight to the loss function, which benefits the training process. This modified loss function is listed in Equation 3:

$$L_{cce}(\mathbf{a}, \mathbf{b}) = -\sum_{j}^{K} a_j \log(b_j),$$

$$L_{cce\_aux}(\mathbf{y}, \mathbf{p}, \mathbf{t}) = -\sum_{i}^{K} (y_i \log(p_i) \times (L_{cce}(\text{sg}[\mathbf{p}], \mathbf{t}) + L_{cce}(\text{sg}[\mathbf{t}], \mathbf{p}))),$$

(3)

where $\mathbf{p}$ and $\mathbf{t}$ represent the output probability vectors of $K$ classes for real and synthetic data, respectively. Additionally, $\mathbf{y}$ denotes the ground truth class probability vector for that sample.

An intuitive understanding of this auxiliary loss is that high-quality synthetic data generated by EEGTrans primarily captures the key characteristics of EEG signals. The BCI Competition IV

Table 4: Ablation on EEGTrans model architecture.

| Subject | EEGTrans (%) | EEGTrans w/o encoder (%) | EEGTrans w/o RVQ autoencoder (%) |
|---|---|---|---|
| Mean | **80.69±7.64** | 53.54±6.86 | 26.19±1.03 |

Dataset 2a was collected some time ago. Unlike the High Gamma Dataset, the experiment may have been conducted without active electromagnetic shielding, leading to significant noise in the data. By having EEGTrans generate synthetic data, we expect this data to primarily reflect EEG signal characteristics, which can serve as a reference to assist the classifier in decision-making. However, since EEGTrans is not trained on the target dataset, the synthetic data likely does not retain subject-specific information, as it has not been exposed to these subjects before. As a result, the performance of models using only synthetic data will inevitably be lower than that of models using only real data. Nonetheless, based on the visual inspection mentioned earlier, it is clear that real and synthetic data closely resemble each other. Therefore, the classifier should yield very similar output probability vectors for these two types of samples. By penalizing the classifier more for significant differences in probability vectors between real and synthetic data, we encourage the model to better align its predictions with the characteristics present in both types of data. A paired t-test with a significance level of $p < 0.05$ was conducted to determine if the performance of the proposed method was significantly better than that of the benchmark. Only EEGTrans showed a significant improvement over the benchmark when the auxiliary loss was applied. Additionally, we performed paired t-tests comparing EEGTrans and CycleGAN using only synthetic data, with EEGTrans significantly outperforming CycleGAN. The results of the High Gamma Dataset can be found in Appendix D.

### 4.8 ABLATION STUDY

We conducted an ablation study on EEGTrans's architecture to assess the encoder-decoder design's impact. Firstly, we removed the encoder, leaving the decoder to generate tokens with the aid of initial ground truth tokens (i.e., 25%) during inference. Secondly, we excluded the RVQ autoencoder, resulting in direct EEG sequence generation by the decoder, trained using mean squared error loss. This adjustment required introducing a zero vector as the substitute for the $SOC$ token during both training and inference phases. Please refer to Appendix E for more details.

The classification results of the ablation study using only synthetic data for training are shown in Table 4. Here, we only present the average classification accuracy across nine subjects. Further details, including individual subject accuracy and data visualization, are available in Appendix E. As shown in Table 4, removing either the encoder or the discrete codes from the proposed framework significantly hinders the model's training effectiveness. In fact, data visualization indicates that without these components, the generated data shows no variation across different channels within the same data or even among different data. If inference starts from the $SOC$ token instead of using 25% ground truth tokens, classification performance is similar to the version without tokens, suggesting that accuracy beyond random guessing is due to the 25% ground truth tokens. During training, EEGTrans without tokens converges with a mean squared error loss of around 1e-6. However, during inference, providing only a zero vector for autoregressive data generation leads to poor performance, indicating that the model ends up learning a trivial solution without tokens.

## 5 CONCLUSION

This paper presents EEGTrans, a framework designed to generate synthetic data for various datasets. By leveraging a transformer-based encoder-decoder architecture and integrating discrete codes into the training process, our model can generalize across multiple datasets. This method produces high-quality synthetic data that enhances downstream classification tasks.

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

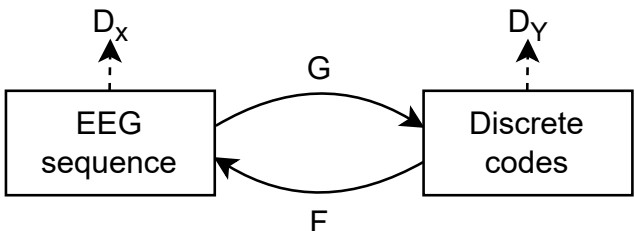

Figure 5: CycleGAN framework. Two generators, $G$ and $F$, are constructed to convert EEG sequences to discrete codes and vice versa. Meanwhile, two discriminators, $D_X$ and $D_Y$, are employed to differentiate between real and synthetic data corresponding to the source and target domains, respectively.

## A  GENERATIVE MODELS

### A.1  CYCLEGAN

CycleGAN (Zhu et al., 2017) was initially developed to translate images from a source domain to a target domain. CycleGAN's framework effectively aligns with our proposed approach, in which EEG sequences serve as the source domain and discrete codes as the target domain. Thus, we incorporate CycleGAN into our proposed framework to compare it with EEGTrans. As depicted in Figure 5, our CycleGAN architecture remains unchanged from the original design, except that we now input EEG sequences and their corresponding discrete codes. However, because both inputs have different dimensions, we cannot employ the identity mapping loss in our case. The training loss for CycleGAN comprises both the adversarial loss and the cycle consistency loss. For a fair comparison, we additionally fine-tune the RVQ decoder during CycleGAN training and utilize it to recover the signals $\hat{x}$ from the generated discrete codes $G(x)$, following the same procedure outlined in EEGTrans.

## B  DATASET

### B.1  DATASET DESCRIPTION

**BCI Competition II Dataset III**  This dataset was collected from a healthy 25-year-old female subject. The task involved controlling a feedback bar using imagery of left or right hand movements. The experiment included 7 runs, each with 40 trials, resulting in a total of 280 trials, each lasting 9 seconds. Data was recorded using a G.tec amplifier and Ag/AgCl electrodes. Three bipolar EEG channels were measured over C3, Cz, and C4. The EEG was sampled at 128Hz and filtered between 0.5 and 30Hz.

**BCI Competition IV Dataset 2b**  This dataset contained EEG data from nine subjects. EEG signals were recorded from three channels (C3, Cz, and C4) at a sampling rate of 250Hz. The data was bandpass-filtered between 0.5Hz and 100Hz, with a notch-filter at 50Hz applied. The cue-based screening involved two classes: motor imagery (MI) of the left hand and right hand. Each subject completed two screening sessions without feedback. Each session included six runs, with ten trials per run and two types of imagery per trial, resulting in 20 trials per run and 120 trials per session. During three online feedback sessions, four runs with smiley feedback were recorded, with each run containing twenty trials for each type of motor imagery. This setup ideally resulted in a total of 720 trials recorded per subject.

**BCI Competition IV Dataset 1**  This dataset was collected from seven healthy subjects who performed motor imagery without feedback throughout the sessions. Each subject was asked to select two motor imagery tasks from three options: left hand, right hand, and foot (side chosen by the subject; optionally both feet). Each trial lasted for a duration of 6 seconds, with a total of 200 trials conducted for each subject for the calibration sessions. The EEG recording was conducted using BrainAmp MR plus amplifiers and an Ag/AgCl electrode cap. Signals from 59 EEG positions were

measured that were most densely distributed over sensorimotor areas. Signals were band-pass filtered between 0.05 and 200 Hz and then digitized at 1000 Hz with 16 bit (0.1 uV) accuracy. The channels' locations were designated as follows: AF3, AF4, F5, F3, F1, Fz, F2, F4, F6, FC5, FC3, FC1, FCz, FC2, FC4, FC6, CFC7, CFC5, CFC3, CFC1, CFC2, CFC4, CFC6, CFC8, T7, C5, C3, C1, Cz, C2, C4, C6, T8, CCP7, CCP5, CCP3, CCP1, CCP2, CCP4, CCP6, CCP8, CP5, CP3, CP1, CPz, CP2, CP4, CP6, P5, P3, P1, Pz, P2, P4, P6, PO1, PO2, O1, O2.

**BCI Competition IV Dataset 2a** This dataset comprises EEG recordings from 9 subjects. The cue-based BCI paradigm involved four different motor imagery tasks: imagining the movement of the left hand, right hand, both feet and tongue. Each subject participated in two recording sessions, resulting in a total of 288 trials per session, with each trial lasting 6 seconds. EEG signals were captured using twenty-two Ag/AgCl electrodes, recorded monopolarly with the left mastoid as reference and the right mastoid as ground. Sampling was done at 250Hz with bandpass filtering applied between 0.5Hz and 100Hz. The amplifier sensitivity was set to 100μV, and a 50Hz notch filter was activated to reduce line noise. The channels' locations were listed as follows: Fz, FC3, FC1, FCz, FC2, FC4, C5, C3, C1, Cz, C2, C4, C6, CP3, CP1, CPz, CP2, CP4, P1, Pz, P2, POz.

**High Gamma Dataset** This dataset comprises data from 14 healthy individuals, each recorded with 128 electrodes. It includes approximately 1000 four-second trials of executed movements spread across 13 runs per subject. The movements fall into four categories: left hand, right hand, both feet, and rest (no movement, but with the same visual cue as the other categories). The training set consists of around 880 trials from all runs except the last two runs, while the test set comprises roughly 160 trials from the last two runs. The recordings were done at a sampling rate of 5 kHz and then resampled to 500 Hz. The recording channels included Fp1, Fp2, Fpz, F7, F3, Fz, F4, F8, FC5, FC1, FC2, FC6, M1, T7, C3, Cz, C4, T8, M2, CP5, CP1, CP2, CP6, P7, P3, Pz, P4, P8, POz, O1, Oz, O2, AF7, AF3, AF4, AF8, F5, F1, F2, F6, FC3, FCz, FC4, C5, C1, C2, C6, CP3, CPz, CP4, P5, P1, P2, P6, PO5, PO3, PO4, PO6, FT7, FT8, TP7, TP8, PO7, PO8, FT9, FT10, TPP9h, TPP10h, PO9, PO10, P9, P10, AFF1, AFz, AFF2, FFC5h, FFC3h, FFC4h, FFC6h, FCC5h, FCC3h, FCC4h, FCC6h, CCP5h, CCP3h, CCP4h, CCP6h, CPP5h, CPP3h, CPP4h, CPP6h, PPO1, PPO2, I1, Iz, I2, AFp3h, AFp4h, AFF5h, AFF6h, FFT7h, FFC1h, FFC2h, FFT8h, FTT9h, FTT7h, FCC1h, FCC2h, FTT8h, FTT10h, TTP7h, CCP1h, CCP2h, TTP8h, TPP7h, CPP1h, CPP2h, TPP8h, PPO9h, PPO5h, PPO6h, PPO10h, POO9h, POO3h, POO4h, POO10h, OI1h, OI2h.

## B.2 Data Preprocessing

The BCI Competition IV Dataset 1 and the High Gamma Dataset each feature numerous channels, some of which are not relevant to the motor imagery task or occur too infrequently for practical use. While a higher number of channels could improve downstream classification tasks, an excess might challenge generative models in distinguishing motor imagery-related signals from unrelated ones. As a result, we excluded such channels during the data preprocessing step. Below are the channels that remained in both datasets after manual selection.

**BCI Competition IV Dataset 1** AF3, AF4, F5, F3, F1, Fz, F2, F4, F6, FC5, FC3, FC1, FCz, FC2, FC4, FC6, C5, C3, C1, Cz, C2, C4, C6, CP5, CP3, CP1, CPz, CP2, CP4, CP6, P5, P3, P1, Pz, P2, P4, P6, PO1, PO2, O1, O2

**High Gamma Dataset** FC5, FC1, FC2, FC6, C3, Cz, C4, CP5, CP1, CP2, CP6, FC3, FCz, FC4, C5, C1, C2, C6, CP3, CPz, CP4, FFC5h, FFC3h, FFC4h, FFC6h, FCC5h, FCC3h, FCC4h, FCC6h, CCP5h, CCP3h, CCP4h, CCP6h, CPP5h, CPP3h, CPP4h, CPP6h, FFC1h, FFC2h, FCC1h, FCC2h, CCP1h, CCP2h, CPP1h, CPP2h

## C Implementation Details

### C.1 RVQ Autoencoder Model Architecture

Here, we provide the details of the RVQ Autoencoder, including the hyperparameters used for each layer.

RVQ autoencoder consists of an encoder, RVQ, and a decoder. RVQ comprises multiple stages that cascade $N_q$ layers of VQ, with each layer containing $k$ codebooks initialized uniformly with dimension $d$. To make it easier to train generative models, the latent variables that represent the codebooks for each layer are shared, which helps in reducing complexity. Additionally, to enhance code utilization, $\ell_2$-normalized codes are used (Yu et al., 2022), and the method employed in SoundStream, which replaces codes with hits below a certain threshold (we set the threshold to 2) with randomly selected vectors from the current batch, is applied. Throughout this study, we fix $N_q$ at 4, $k$ at 2048, and $d$ at 128 for the RVQ autoencoder. We choose these values as they provide high code utilization while maintaining a low reconstruction loss, and set $\beta$ to 10 in the training loss. The RVQ encoder's configuration will ultimately determine the timestamps within the discrete token space ($\tau$). We set the compression factor to be 4 ($\tau = \frac{T}{4}$).

The encoder features several 1D convolutional layers and GELU activations. It starts with a Conv1d layer (1 input channel, 16 output channels, kernel size 3, stride 1, padding 1, no bias), followed by GELU. Next is another Conv1d (16 to 32 channels, kernel size 3, stride 1, padding 1, no bias), a grouped downsampling Conv1d (32 to 32 channels, kernel size 2, stride 2, groups 32, no bias), and GELU. This is followed by a Conv1d (32 to 64 channels, kernel size 3, stride 1, padding 1, no bias) and GELU. The final layers include a Conv1d (64 to 128 channels, kernel size 3, stride 1, padding 1, no bias) and another grouped downsampling Conv1d (128 to 128 channels, kernel size 2, stride 2, groups 128, no bias).

The decoder features a combination of 1D convolutional and transposed convolutional layers along with GELU activations. It starts with a transposed convolution (128 input and output channels, kernel size 2, stride 2, groups 128, no bias), followed by a Conv1d (128 to 64 channels, kernel size 3, stride 1, padding 1, no bias), and GELU. Next, it includes a Conv1d (64 to 32 channels, kernel size 3, stride 1, padding 1, no bias) with GELU, followed by a transposed convolution (32 input and output channels, kernel size 2, stride 2, groups 32, no bias). This is followed by a Conv1d (32 to 16 channels, kernel size 3, stride 1, padding 1, no bias) with GELU, and the final layer is a Conv1d (16 to 1 channel, kernel size 3, stride 1, padding 1, no bias).

## C.2 RVQ CODEBOOK UTILIZATION

We ran two versions of the RVQ autoencoder by varying the number of codebooks, while keeping the encoder and decoder architecture unchanged, to examine the effect of codebook number on the quantization process. We trained the autoencoder with 2048 codebooks (as in previous works like DeWave (Duan et al., 2023)) and a larger set of 16384 codebooks. With 2048 codebooks, the mean squared error (MSE) loss between the EEG sequence and its reconstructed sequence in the BCI Competition IV Dataset 2a was 0.152, while with 16384 codebooks, the MSE loss was 0.131. Although more codebooks improve quantization performance, they significantly increase memory consumption because each latent variable requires GPU memory allocation, with 16384 codebooks requiring more than 24GB of GPU memory during training. Another metric, active code (code utilization), showed that roughly 50% of the codes were used with 16384 codebooks, compared to almost 100% utilization with 2048 codebooks. Therefore, we decided to use the 2048 codebook version.

## C.3 EEGTRANS MODEL ARCHITECTURE

The architecture of the proposed EEGTrans utilized in this study is depicted in Figure 2. EEGTrans includes both an encoder and a decoder. In the encoder, the input embedding layer is made up of 6 Conv2D layers, which closely resemble the encoder of the RVQ autoencoder, except for the fact that the last two convolutional layers have output channels of 256. The encoder block consists of 4 layers of attention blocks with an embedding size of 256 and 4 attention heads. Similarly, the decoder block follows the same configuration as the encoder block, except for the input embedding layer, which is a simple lookup table storing embeddings of a fixed dictionary size of 256. Subsequently, the output undergoes processing through a MLP with two linear layers of output dimensions 256 and 2048, incorporating a GELU activation function in between, to map the embeddings to discrete tokens.

### C.4  CYCLEGAN MODEL ARCHITECTURE

As shown in Figure 5, CycleGAN consists of two generators $G$ and $F$ and two discriminators $D_x$ and $D_y$.

Generator $G$ is composed of convolutional layers and a MLP. These convolutional layers fall into two categories: those with padding and those without. The ones with padding start with a ReflectionPad1d layer (padding 1), followed by a Conv1d layer (kernel size 3, stride 1, padding 0), InstanceNorm1d layers, and a GELU activation function. Conversely, the ones without padding begin with a Conv1d layer (kernel size 2, stride 2, padding 0), followed by InstanceNorm1d layers, and a GELU activation function. The arrangement of this generator follows a pattern of two padding convolutional blocks, one downsampling convolutional block, two padding convolutional blocks, one downsampling convolutional block, and finally, a MLP containing two linear layers with output dimensions of 128 and 2048, with a GELU activation function between them, to map the embeddings to discrete tokens.

Generator $F$ is composed of transpose convolutional layers and convolutional layers, basically a reverse process of Generator $G$. The transpose convolutional block begins with a ConvTranspose1d layer (with a kernel size of 2 and a stride of 2), succeeded by a ReflectionPad1d layer (with a padding of 1), a Conv1d layer (with a kernel size of 3, a stride of 1, and no padding), InstanceNorm1d layers, and finally, a GELU activation function. The convolutional block comprises a ReflectionPad1d layer (with a padding of 1) and a Conv1d layer (with a kernel size of 3, a stride of 1, and no padding). To reconstruct a continuous signal from discrete codes, this generator is built with three transpose convolutional blocks followed by one convolutional block. A token embedding is also required to convert the inputs from discrete codes into vectors.

Discriminator $D_x$ is composed of a sequence of Conv1d layers. Initially, there is a Conv1d layer with a kernel size of 4, a stride of 2, padding of 1, and no bias, followed by a LeakyReLU activation function. Subsequently, three consecutive convolutional blocks consist of a Conv1d layer (kernel size 4, stride 2, padding 1, no bias), an InstanceNorm1d layer, and a LeakyReLU activation function. The last Conv1d layer in these blocks has a stride of 1. Finally, the model concludes with a Conv1d layer (kernel size 4, stride 1, padding 1, no bias). The model's output undergoes average pooling in the timestamp dimension to distinguish whether the input data is real or synthetic. Discriminator $D_y$ shares a similar model architecture with Discriminator $D_y$, with the distinction that it operates with discrete input. Therefore, a token embedding is necessary to convert the code into vectors.

### C.5  TRAINING AND INFERENCE OF GENERATIVE MODELS

The training procedure for EEGTrans is outlined in Section 4.3, but here are some additional details. We train EEGTrans for 1000 epochs, although overfitting to the source datasets typically starts after about 100 epochs. To address this, we employ an early stopping technique. Additionally, we select the model checkpoint that performs best on the target datasets by monitoring the cross-entropy loss of these datasets, which is then used for inference. With early stopping, the training typically takes less than a day on a single RTX 4090 GPU.

CycleGAN is trained using the Adam optimizer and the AdamW optimizer, respectively, with the same learning rate scheduler as EEGTrans. We apply the same early stopping strategy for Cycle-GAN. However, for CycleGAN, the checkpoints selected for inference are based on the smallest generator loss on the target datasets. Utilizing the early stopping strategy enables us to maintain the training duration under a day.

## D  CLASSIFICATION TASK

We adhered to the original implementation of EEGNet (Lawhern et al., 2018) and implemented it on these two datasets. However, we opted to eliminate the max norm constraint on the Dense layer, as we observed that its removal can lead to slight performance improvements, particularly during longer training periods. Table 5 provides comprehensive details regarding the architecture of EEGNet. In all experiments, EEGNet is trained individually for each subject, employing the Adam optimizer with a learning rate of 1e-3. Training occurs over 1000 epochs utilizing categorical cross-entropy loss.

Table 5: EEGNet architecture details. Conv2D includes batch normalization; DepthwiseConv2D and SeparableConv2D include batch normalization and ELU activation function; AveragePooling2D includes dropout regularization with a rate of 0.5. Dense includes softmax activation function. C denotes channels, which are 22 for the BCI Competition IV Dataset 2a and 45 for the High Gamma Dataset. Following the original implementation, we also regularize each spatial filter by using a maximum norm constraint of 1 on the weights of the DepthwiseConv2D.

| Name | Layer | Filters | Depth | Kernel | Padding | Output shape |
|------|-------|---------|-------|--------|---------|--------------|
| C1 | Conv2D | 16 | - | $1\times64$ | Same | $C\times256\times16$ |
| DC1 | Depthwise-Conv2D | - | 2 | $C\times1$ | Valid | $1\times256\times32$ |
| AP1 | Average-Pooling2D | - | - | $1\times4$ | - | $1\times64\times32$ |
| S1 | Separable-Conv2D | 32 | 1 | $1\times16$ | Same | $1\times64\times32$ |
| AP2 | Average-Pooling2D | - | - | $1\times8$ | - | $1\times8\times32$ |
| F1 | Flatten | - | - | - | - | 256 |
| D1 | Dense | - | - | - | - | 4 |

Table 6: Classification performance on High Gamma Dataset. In this table, "R" denotes using only real data, "S" denotes using only synthetic data, "RS" stands for combining real and synthetic data, and "Aux" signifies combining real and synthetic data with auxiliary loss. While the inclusion of synthetic data in this dataset does not significantly boost classification accuracy, EEGTrans still outperforms CycleGAN under same conditions, demonstrating its effectiveness in generating synthetic data across various datasets.

| | | EEGTrans (%) | | | CycleGAN (%) | | |
|---|---|---|---|---|---|---|---|
| Subject | R | S | RS | Aux | S | RS | Aux |
| 1 | 91.87 | 92.29 | 91.87 | **93.33** | 73.54 | 89.58 | 90.83 |
| 2 | 88.28 | 89.00 | 87.77 | 90.03 | 70.81 | 87.77 | **90.33** |
| 3 | **93.94** | 92.59 | 92.98 | 93.26 | 76.05 | 90.28 | 93.17 |
| 4 | 94.41 | 90.82 | 94.70 | **95.27** | 77.38 | 92.99 | 93.84 |
| 5 | 91.81 | 89.20 | **92.72** | 92.38 | 73.97 | 88.18 | 90.68 |
| 6 | 88.17 | 87.98 | **89.13** | 87.88 | 77.11 | 84.71 | 88.55 |
| 7 | 92.30 | 92.40 | 91.92 | 91.53 | 66.53 | 90.09 | **92.69** |
| 8 | 92.50 | 91.76 | **92.75** | 92.13 | 52.46 | 88.69 | 91.76 |
| 9 | 90.67 | **92.50** | 91.92 | 91.25 | 78.65 | 89.03 | 90.28 |
| 10 | 84.90 | 85.19 | **86.25** | 83.46 | 51.05 | 82.50 | 84.23 |
| 11 | 77.98 | **78.55** | 77.11 | 78.46 | 63.07 | 75.86 | 77.78 |
| 12 | **95.76** | 91.73 | 94.61 | 94.90 | 53.07 | 91.92 | 92.30 |
| 13 | 91.35 | 90.93 | 91.66 | **92.29** | 67.08 | 89.06 | 90.83 |
| 14 | 94.32 | 93.26 | 95.00 | **95.67** | 61.53 | 91.92 | **95.67** |
| Mean | 90.59 | 89.87 | 90.74 | **90.84** | 67.31 | 88.04 | 90.21 |

Table 7: Ablation on EEGTrans model architecture.

| Subject | EEGTrans (%) | EEGTrans w/o encoder (%) | EEGTrans w/o RVQ autoencoder (%) |
|---------|--------------|--------------------------|----------------------------------|
| 1 | **85.59±3.64** | 49.82±1.85 | 24.82±2.02 |
| 2 | **64.59±3.13** | 44.10±4.76 | 27.77±3.04 |
| 3 | **87.14±4.32** | 53.64±2.42 | 25.17±3.74 |
| 4 | **74.82±3.35** | 52.06±6.28 | 26.04±2.28 |
| 5 | **81.77±2.71** | 65.62±5.11 | 25.17±5.37 |
| 6 | **74.99±4.38** | 48.94±4.55 | 26.21±1.66 |
| 7 | **83.86±3.27** | 62.50±5.08 | 25.87±3.42 |
| 8 | **81.76±1.85** | 46.68±4.57 | 27.77±2.56 |
| 9 | **91.67±1.14** | 58.49±5.47 | 26.91±1.69 |
| Mean | **80.69±7.64** | 53.54±6.86 | 26.19±1.03 |

The classification results of the High Gamma Dataset are presented in Table 6. Using the same experimental settings as in BCI Competition IV Dataset 2a, we will report the five-fold cross-validation mean accuracy for each subject. Although incorporating synthetic data into the training process does not significantly enhance classification performance in this dataset, it is worth noting that for 6 subjects, using only synthetic data outperforms using only real data. Additionally, the average performance gap between subjects is not as large as it was in the previous dataset. Since the High Gamma Dataset was acquired in an EEG lab with a technical setup that included active electromagnetic shielding, and subjects sat in a comfortable armchair inside a dimly lit Faraday cabin, the collected data is less susceptible to noise. Therefore, the real data predominantly reflects true motor imagery EEG characteristics. Even when EEGTrans generates synthetic data, the synthetic data may possess similar features, resulting in performance improvement that is not comparable to that of the BCI Competition IV Dataset 2a. Nonetheless, EEGTrans continues to perform better in generating high-quality synthetic data and outperforms CycleGAN.

# E  ABLATION STUDY

We performed an ablation study on the architecture of EEGTrans to determine the impact of the encoder-decoder design on the final results. Additionally, we tested the model without the RVQ autoencoder to see if it could still deliver satisfactory performance. First, we remove the encoder architecture from EEGTrans while keeping everything else unchanged. This means the decoder can no longer use cross-attention on the EEG sequences when generating discrete tokens. Since the generated data would be completely random without an encoder if we start the inference from the $SOC$ token, we address this by providing the first 25% of the ground truth discrete tokens as inputs to the decoder during inference. Second, we exclude the RVQ autoencoder from the framework, so the decoder directly generates the continuous EEG sequence. Thus, the decoder is now trained using mean squared error loss, minimizing the distance between the generated synthetic data and the corresponding real data, instead of using cross-entropy loss. After removing the RVQ autoencoder, there is no $SOC$ token anymore. Therefore, the model is trained with a vector consisting of all zeros prepended at the front, which serves as the $SOC$ token in continuous form. During inference, only this zero vector is provided initially for the decoder.

The classification results of the ablation study using only synthetic data for training are shown in Table 7. The synthetic data generated by EEGTrans, without the encoder and RVQ autoencoder, is shown in Figures 6 and 7. In Figure 6, it is evident that only the segments where ground truth tokens are provided closely resemble real data. When EEGTrans starts generating tokens autoregressively, the synthetic data lacks meaningful EEG features. This is reflected in the training loss, indicating that EEGTrans does not train well without the encoder. Therefore, the encoder currently plays a crucial role in EEGTrans. However, one limitation that needs to be addressed in the future is the requirement for entire EEG sequences as input for the encoder. As shown in Figure 7, the amplitude of the synthetic data is nearly zero during inference.

If the inference starts from the $SOC$ token instead of using 25% ground truth tokens, the classification performance would be similar to the version without tokens. This suggests that the accuracy

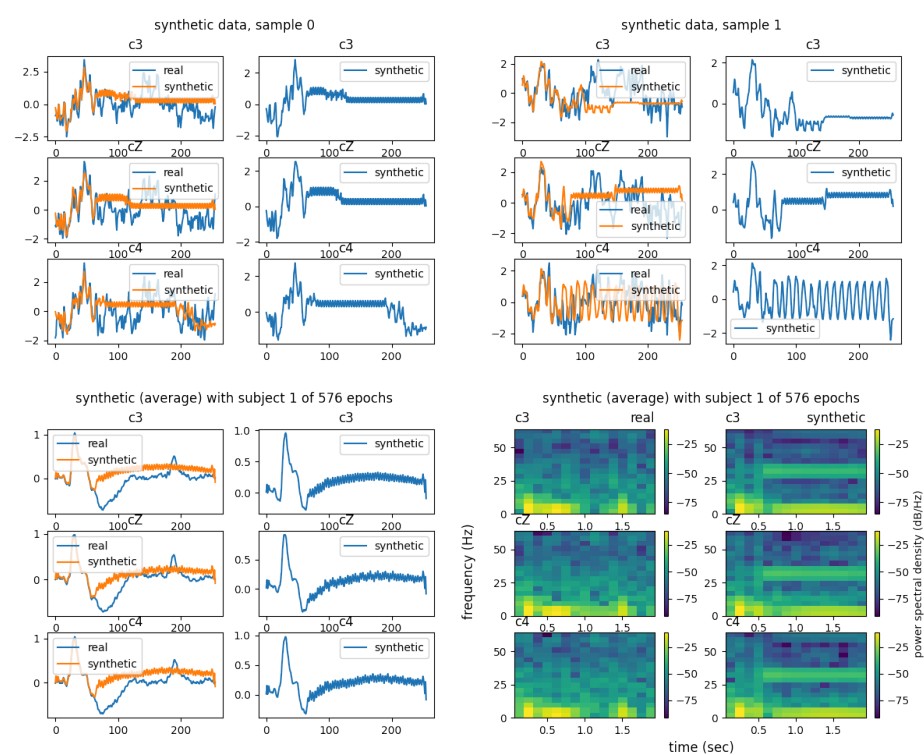

Figure 6: Visualization of synthetic data generated by EEGTrans without encoder for Subject 1 in the BCI Competition IV Dataset 2a.

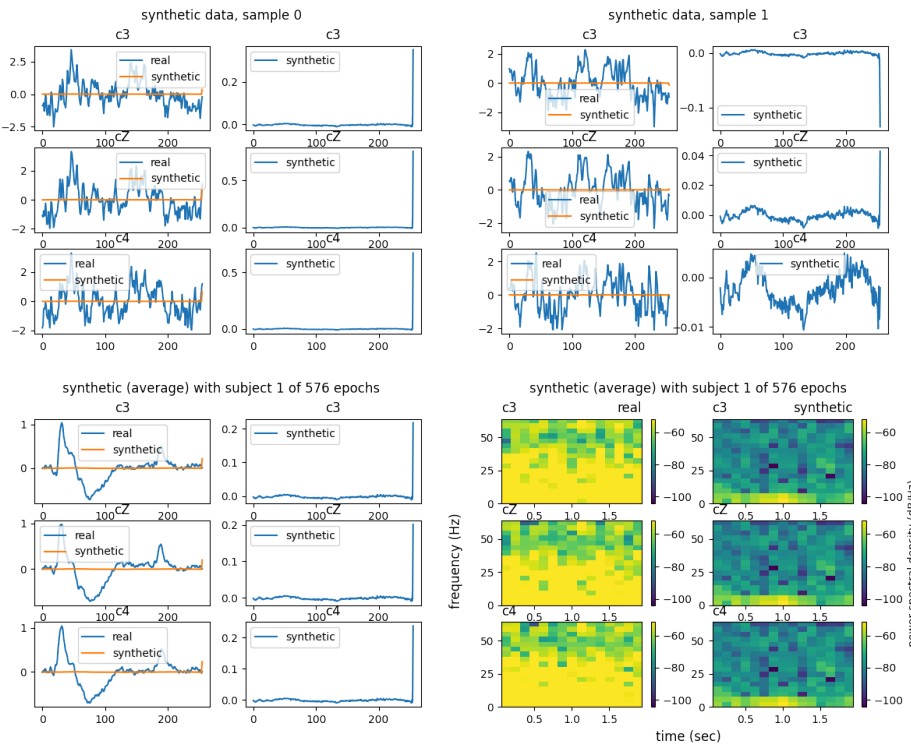

Figure 7: Visualization of synthetic data generated by EEGTrans without RVQ autoencoder for Subject 1 in the BCI Competition IV Dataset 2a.

exceeding random guessing is entirely due to the inclusion of the 25% ground truth tokens. On the other hand, during the training process, EEGTrans without tokens performs very well in predicting the next timestamp, achieving a mean squared error loss on the scale of 1e-6. This indicates that the model has already converged on the training task. However, due to the high temporal proximity of EEG signals and the availability of ground truth signals up to the current timestamp during training, the model can achieve good predictions by simply replicating the current timestamp value or learning the difference between the next timestamp and the current one, then adding this difference to the current timestamp to predict the next value. However, only a zero vector is provided during inference, causing the model to perform poorly. This indicates that without tokens, the model has learned a trivial solution. In fact, if we run inference using teacher-forcing settings, providing signals up to the current timestamp when predicting the next one, the synthetic data closely resembles the real data.

