# Rebuttal Figures for "EEGTrans: Transformer-Driven Generative Models for EEG Synthesis"

We sincerely appreciate the time and effort the reviewers and ACs have dedicated to evaluating our submission. We have made every effort to address all the concerns raised and hope that our responses provide clarity. This document includes all the figures referenced in our rebuttal. The detailed responses can be found in the text boxes on the OpenReview website.

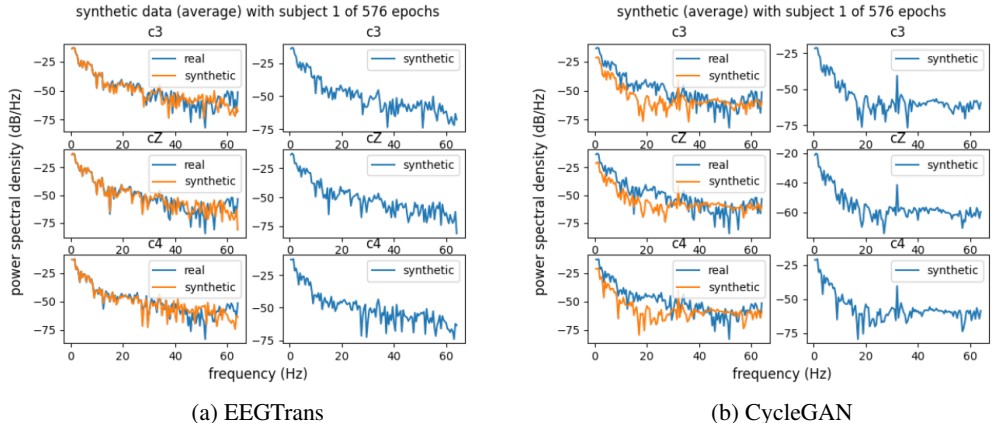

(a) EEGTrans                                      (b) CycleGAN

Figure 1: Power Spectral Density Comparison for Synthetic Data Generated by EEGTrans and CycleGAN. The plots display the power spectral density of evoked data (averaged over 576 epochs) for subject 1 across c3, cZ and c4 channels. (a) For EEGTrans, the PSD of synthetic data closely aligns with real data in the lower frequency range (below 30 Hz), though it gradually diverges in higher frequencies. Crucially, the key motor imagery frequency bands are well-preserved. (b) In contrast, CycleGAN-generated synthetic data shows significant deviations from real data across all frequency ranges.

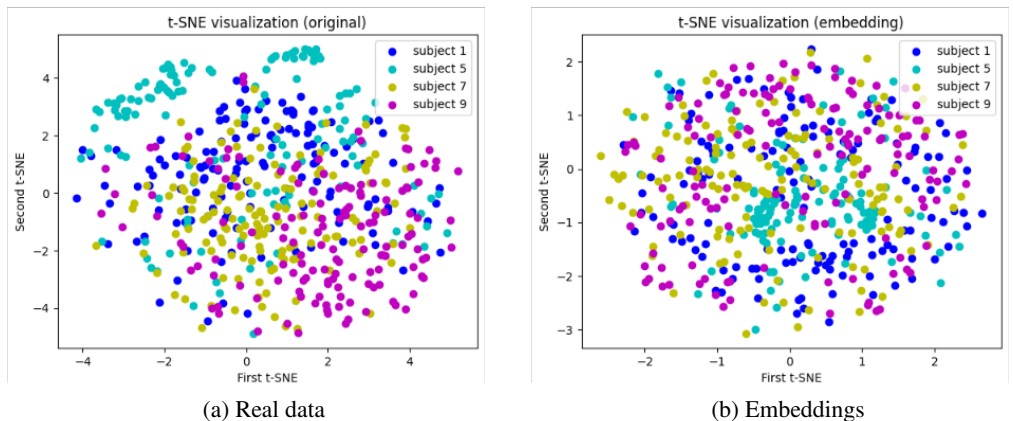

(a) Real data                                      (b) Embeddings

Figure 2: t-SNE Visualization Comparison. This figure contrasts the representations of real data with those generated by an encoder using embeddings. Data from four randomly selected subjects (subjects 1, 5, 7, and 9) are visualized. (a) In the original data, distinct clustering patterns emerge (e.g., subject 5 clusters in the top-left, while subject 9 clusters in the bottom-right). (b) In contrast, the embeddings show a more dispersed distribution, with most subjects' data points spreading outward from the center.

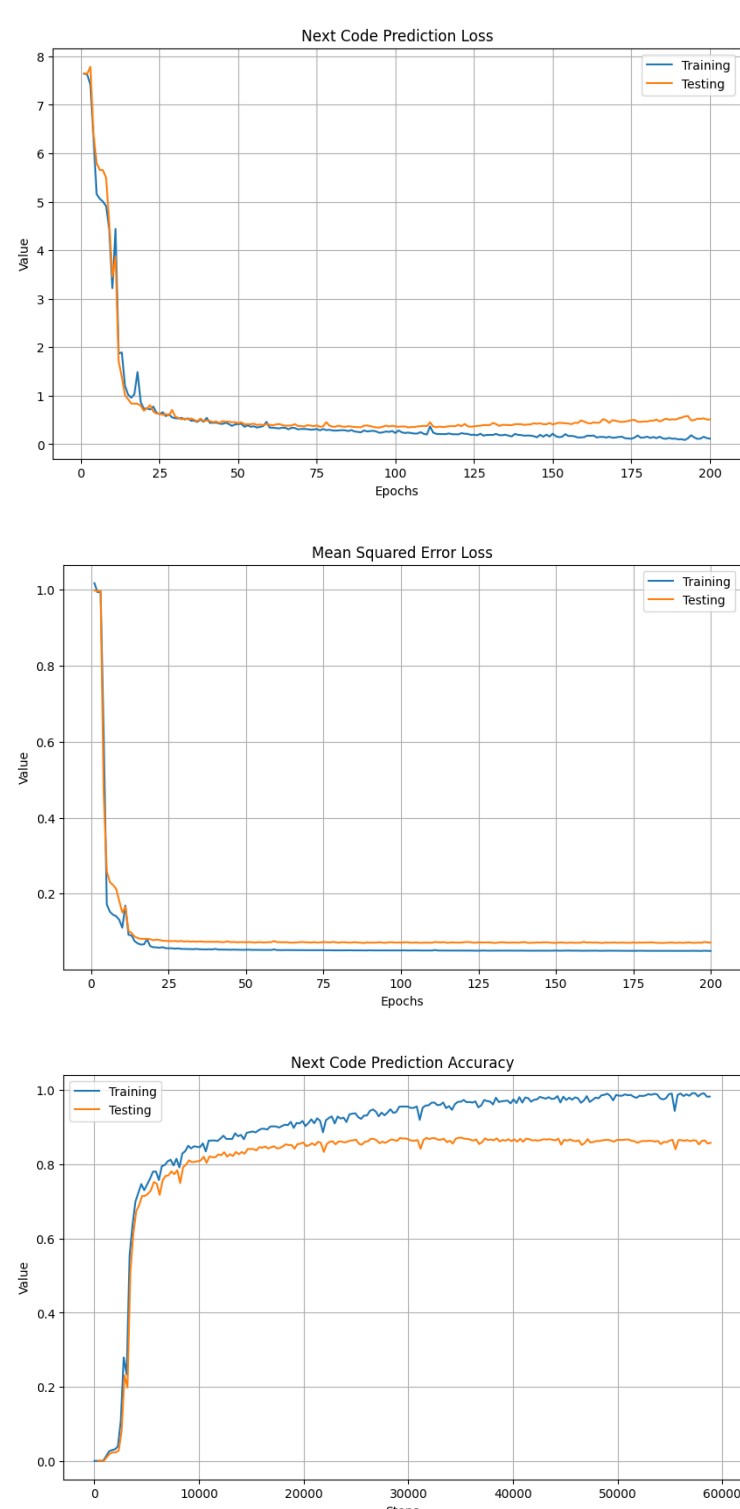

Figure 3: Training and Testing Loss and Accuracy Curves. The plots depict the training and testing performance metrics during the training of EEGTrans. The top graph shows the next code prediction loss across epochs, while the middle graph presents the mean squared error loss (fine-tuning RVQ decoder) over the same epochs. The bottom graph illustrates the next code prediction accuracy as a function of training steps. The results highlight the convergence behavior and the generalization capability of the model.