# OpenReview forum: "EEGTrans: Transformer-Driven Generative Models for EEG Synthesis"
_ICLR.cc/2025/Conference — Submitted to ICLR 2025_

### Official Review · Reviewer_v37w · 2024-10-31

**Soundness:** 2
**Presentation:** 3
**Contribution:** 2
**Rating:** 3
**Confidence:** 4

**Summary:**

The paper proposed a transformer-based generative model, EEGTrans to autoregressively generate EEG data based on the input sequence. The proposed model was evaluated against CycleGAN and achieved more accurate 'prediction' or generation of future segments.

**Strengths:**

1. the structure of the paper is good and clear
2. the method section was clearly described and the figure 2 is informative.
3. the source codes are provided to facilitate reproduction

**Weaknesses:**

1. The model was trained for next code prediction which means the proposed method is more of a EEG signal 'forecasting' or 'prediction' rather than generation for data augmentation purpose. Therefore, the introduction and problem setup is miss aligned.
2. For synthetic data generation to facilitate training in a new dataset, both the reality and the diversity of generated signals are important. In a typical GAN or Diffusion setting, new signals can be generated or sampled from a random vector sampled from the normal distribution during inference to achieve diversity and variety. However, in the proposed method, it seems like there is no sampling process for diverse signal generation, so how can the proposed method augment the target dataset?
3. There are GAN-based method for EEG that can directly generate the raw signal which should be added into comparison. For example, EEG-GAN.
4.The language used in section 4.7 is very vague, for example, 'Dataset 2a was collected some time ago', 'the synthetic data likely does not retain subject-specific information', 'the classifier should yield very similar output'. These claims need to be justified better for example, providing a tSNE plot to show the generated data vs the real data, and the target classes vs the subject to demonstrate there is no subject-specific information.
5. Why not use MSE as a performance metric since the method is 'predicting' the future segment?
6. The use of GAN to replace the Transformer module as baseline for next coder prediction task should be justified better.

**Questions:**

1. The EEG signals have high variations between subjects and between datasets, how are these addressed in this work? Is the generation subject-dependent?

2. The generation quality of the proposed will also be largely dependent on the RVQ AE, which added additional complexity to train and tune, any comment on this?

3. What are the real-world applications for the proposed method? It seems like in-order to generate synthetic data for a new target dataset, you still need to use real data as input. It is very rare that we only collect 'the first half of a trial' in BCI experiments.

4. For auto-regressive generation, how does the 'prediction' performance decline with respect of length?

---

### Official Review · Reviewer_DaYo · 2024-11-01

**Soundness:** 3
**Presentation:** 3
**Contribution:** 2
**Rating:** 5
**Confidence:** 5

**Summary:**

This paper proposes the transformer-based EEG generative model, namely EEGTrans, to synthesize EEG data based on multiple datasets. The model used the RVQ autoencoder to train the discretize the EEG signal and build the codebook to represent EEG tokens. Then EEGTrans can model the generative task as a next-token prediction task in an auto-regressive manner just as what a language model is doing and then the RVQ decoder can generate the EEG signal similar to the real data. Experiments confirms the similarity by comparing the spectral entropy and sample entropy. Further experiments confirms that with these synthetic data as augmentation, a model can achieve better results on the target BCI dataset.

**Strengths:**

1. The general idea is taken from the language model, thus the overall methodology is reasonably backed up with empirical results, and is still novel in the EEG synth task.
2. The experiments demonstrate their superior performance compared to a classic method CycleGAN.
3. The overall presentation is good. The authors clearly conveys their ideas.

**Weaknesses:**

First of all, I have to mention that the key research problem of the EEG synthesis is not "what is the best method to generate the synthetic EEG data", but "why do we need to generate the synthetic EEG data". The paper answers this question in the most common way, which is to use these data as augmented data to improve the performance on a specific downstream task (and in this paper the BCI motor imagery task). From this point of view, the scope of the contribution of this paper will be limited to getting improved synthetic EEG data for data augmentation rather than extending our understanding for this area.

The above statement is actually not the weakness of this paper. I like paper that won't overclaim but focus more on concrete things. However, starting from it, there will be the following concerns about the paper.
1. It's unclear to me whether the label needs to be generated together with the EEG data. If so, the usage of the synthetic dataset seems to be limited to highly related tasks and the claimed cross-dataset advantage will be weakened.
2. If the synthetic data is used to boost the performance of downstream tasks, then how does it perform compared to more common but simple data augmentation methods like cropping and concatenating EEG fragments, jittering, etc.? It lacks such justification.
3. If the synthetic data is claimed to integrate information from different datasets, then how should we compare this paradigm to a more common paradigm where model such as LaBraM will integrate information in the hidden representation space? This alternative paradigm can boost the performance of downstream tasks with unsupervised training on unlabeled EEG data, which can integrate information from more diverse dataset.

There are also some comments on the methodology:
1. EEGTrans doesn't take the relationship among channels into consideration, and doesn't evaluate the cross-channel reconstruction quality.
2. The role that the next-token prediction plays in the proposed method is unclear and unverified. See the questions for more details.

**Questions:**

1. Does the next-token prediction converge well? I'm curious about how well the next-token prediction would work. Do you have a test set to verify the next-token prediction process? If there is the test set, how well can the model correctly predict the next token? Maybe reporting metrics like perplexity will help.
2. Following Question 1., it's possible that the model is simply yielding EEG data from the trained dataset (with some kind of bias on frequency as described in 4.5). If so, the model will be downgraded to merely replicate an existing dataset and use it to other datasets.
3. While the model is called EEGTrans, taken the name from transformer, I feel the RVQ plays the most important role for synthesizing realistic EEG data. So what if we use the discretization method such as RVQ, but do not use the auto-regressive manner to train a transformer model? For example, one can actually combine VQ with CycleGAN to also generate good-quality EEG data.
4. I'm not sure whether it is my problem, but when I'm trying to visit the link to the code in the abstract, it tells me that "The repository is not found."

---

### Official Review · Reviewer_tVd8 · 2024-11-01

**Soundness:** 1
**Presentation:** 2
**Contribution:** 2
**Rating:** 3
**Confidence:** 5

**Summary:**

The authors proposed EEGTrans (Encoder - Decoder) based architecture for generating synthetic EEG datasets. It is useful, as collecting EEG data is a human subject-dependent and time-consuming process. Authors evaluated on multiple motor-imagery datasets after downsampling them to 128 Hz.

**Strengths:**

The paper used Transformer architecture to generate synthetic EEG. The idea is not novel as there have been previous attempts on this, however the problem is not well studied. Thus utilising transformers seems a reasonable attempt at testing the capability.
They Validated on multiple datasets.

**Weaknesses:**

There have been other methods where authors used GPT-based architecture to claim a foundation Model: "Neuro-GPT: Towards A Foundation Model for EEG". There is another paper "A Time Series is Worth 64 Words: Long-term Forecasting with Transformers" for time series data -  The authors should have evaluated that.

There should be more details on architecture, and optimisation. The authors explained for EEGTrans but for CycleGAN approach, I couldn't find the information on layers/number of parameters/etc. There is some information in the appendix but still not complete.
I would encourage authors to include maybe clear architecture diagram for both in the main text, so it is easier to compare and understand the implementation.

Additionally, There is an extreme weakness in the paper related to evaluating synthetic data. The authors didn't perform standard evaluations like PSD, STFT, activation in alpha and beta bands, or source analysis. Averaging all the subjects doesn't distinguish activations.

There are 5 datasets for evaluation: Only 1 dataset is there with 1 subject and 128 Hz sampling. Why use that dataset to bring the other 4 datasets with higher resolution and number of subjects to 128 Hz? Maybe let go of that dataset and deal with 250 Hz, as we have tried over the years to have higher sampling and temporal resolution for these datasets

In paper, I have seen repetitive occurrences stating the same stuff, authors can reduce that, to add more information in main text. ex:
"Visual inspection indicates that EEGTrans produces higher-quality synthetic data compared to CycleGAN. It is evident from the
visual comparison that EEGTrans’s generated data is significantly superior".

another Line [493]The BCI Competition IV Dataset 2a was collected some time ago] - "some time ago" really? We can say, 2a was collected in 2008 and HGD was in 2017. However the may statement is speculative in nature, so authors can point towards noise/interference due to room/environment.

Line [534] : "This method produces high-quality synthetic data that enhances downstream classification tasks." - If you only intended for classification (only one downstream), just train a classification model utilizing transformers?

**Questions:**

This paper will need a lot of work, to perform all the experiments and evaluate the synthetic EEG Data.

Can the authors perform simple group-level beamformer source analysis for both real and synthetic data?

Additionally Time Frequency plots for beta power changes in case of motor imagery for both generated and real data?

Can we get some loss plots for the training and fine-tuning?

Line [318-320] "We focus on three channels commonly used in motor imagery experiments, and two epochs are shown to allow us to confirm the robust performance of EEGTrans across different channels and multiple epochs." - which 3 channels - c3, c4, cz?

Line [416]"As shown in Table 2, only EEGTrans closely matches the ground truth (real data) with minor differences, retaining high sample entropy and thus indicating high complexity and variation. However, it does not retain high-frequency components, which is evident in the spectral entropy. " - Can we have some statistical comparison?

Line[463]"Table 3 shows the classification accuracy achieved through various approaches: using only real data, only synthetic data, combining real and synthetic data, and incorporating real data with synthetic data along with auxiliary loss. For a comprehensive analysis, we employ five-fold cross-validation instead of the original train-test split used in the competition." - How did you do 5 fold cross- validation? What was your distribution of the test and validation set, or it was simple 80 -20? Since it's already a competition dataset, why did you choose to do cross-validation and not cross-session analysis?

Can you also do inter-subject analysis?

Is it justified to benchmark against cycle gan for non-stationary time series data?

---

### Official Review · Reviewer_yv5f · 2024-11-04

**Soundness:** 2
**Presentation:** 2
**Contribution:** 2
**Rating:** 3
**Confidence:** 3

**Summary:**

This manuscript describes a vector-quantization-based autoregressive transformer as a generative model for EEG. They use their generative model to generate synthetic data for EEG classification tasks. Results show that using the real data and the synthetic data with an auxiliary loss slightly outperforms using only the real data on motor imagery decoding tasks.

**Strengths:**

* applies commonly used, yet fairly recentnovel deep learning methods from other fields to EEG, where they have not been used as much
* supplies code
* overall approach probably makes sense (even though have not fully understood it)

**Weaknesses:**

I found the way how actually the synthetic data is generated hard to understand and may still not have fully understood it. This should be more clearly described early on I am still confused about it.

Some of the writing I found vague and therefore hard to read, e.g. first sentences "Large language models (LLMs) have been extensively utilized across various scenarios due to their powerful model characteristic: the generative models. These models are not restricted to producing specific forms of output; instead, they can generate output in any form" I found these sentences mostly confusing.

It is unclear to me how the authors split the data into training and test, I think they did not follow official train/test splits of the datasets? Did not see this information, maybe I missed it.

Also, there are of course a lot of existing works on those data that should be compared to, and for this it is also necessary to check and align the train/test split with those works and to also mention their performances.

Average power spectra (e.g., for all trials fo one subject) for synthetic and generated data should be shown to assess quality of generated EEG.

**Questions:**

As written, still do not understand how data is exactly generated for the classification tasks. Can you try to explain it very concisely?

Also, what train/test splits did you use? In the different training stages (generative model training, classifier training etc.)

---

### Author Response · Authors · 2024-11-20
**Response 1**

We sincerely thank all the reviewers and ACs for taking the time to thoroughly read our paper and provide valuable comments and feedback. Your insights have been incredibly helpful in refining our work.

Since some of the reviewers raised overlapping questions, we have consolidated our responses into a single section to address all concerns comprehensively.

## Generative Model
In this paper, we focus on developing a generative model capable of generating EEG signals across different modalities (i.e., discrete tokens and continuous signals) and variable lengths, which cannot be achieved using classification-based transformer models (e.g., LaBraM and EEG Conformer). For details on how we generate synthetic data, refer to Section 4.4: “For EEGTrans, synthetic data is created by feeding real data into the encoder, which then prompts the decoder to iteratively generate the next timestamp code starting from the SOC token. The fine-tuned RVQ decoder is then used to convert the discrete codes back into signals.” Please kindly refer to Figure 2 for a visual representation. EEGTrans is provided with real data for the encoder, and the decoder starts generating tokens autoregressively with only the SOC token provided at the beginning. Once all tokens are generated, the RVQ decoder is used to generate the corresponding pair of synthetic EEG signals.

## Training/Test Split
Regarding the training/test split, there are two parts to consider. The first part involves how the dataset is split when training EEGTrans. As listed in Section 4.1, we used three datasets (BCI Competition II Dataset III, BCI Competition IV Dataset 2b, and BCI Competition IV Dataset 1) to train and test on two other datasets (BCI Competition IV Dataset 2a and the High Gamma Dataset). The second part involves evaluating the quality of synthetic data on EEGTrans's test datasets. Since the motor imagery paradigm is originally a classification task, we chose to evaluate synthetic data quality using this classification approach. We performed five-fold cross-validation (as described in Section 4.7) on both test datasets to ensure a more robust evaluation, unlike the original train-test split that only tests the synthetic data **once**. We used stratified k-fold to maintain the same class distribution in both training and testing sets as in the original dataset. We reported the mean and standard deviation of the five-fold accuracy. Moreover, training the classification model (EEGNet) with synthetic data using auxiliary loss showed significant improvements over the baseline (using only real data), as demonstrated by a paired t-test with a p-value < 0.05.

## CycleGAN Architecture
The architecture details of CycleGAN are provided in Appendix C.4. For a complete breakdown, please refer to that section.

## Clarification on Previous Statements
We apologize for any confusion caused by our earlier wording, specifically the phrase "some time ago." What we intended to emphasize was the timing and conditions of data collection. As detailed in Appendix D: "The High Gamma Dataset was acquired in an EEG lab with a technical setup that included active electromagnetic shielding. Participants sat in a comfortable armchair inside a dimly lit Faraday cabin, making the collected data less susceptible to noise." We acknowledge that our initial explanation could have been clearer and appreciate your understanding.

“We focus on three channels commonly used in motor imagery experiments, and two epochs are shown to allow us to confirm the robust performance of EEGTrans across different channels and multiple epochs.” In Figure 3, we provide a detailed comparison of the synthetic data generated by EEGTrans, presenting the real data alongside the corresponding synthetic data for Subject 1. The channels used are listed, and legends are included to differentiate between real and synthetic data. Please refer to Figure 3 for more details.

## Frequency Analysis
We conducted a frequency analysis, which revealed that the synthetic data generated by EEGTrans closely resembles real data in the low-frequency band (below 30 Hz), though it shows slightly lower power in the high-frequency band (above 30 Hz). The frequency plots were highly correlated with those presented in the spectrograms; therefore, we chose to omit the frequency analysis plots in the manuscript to avoid redundancy. However, you can find the power spectral density plot in the supplementary material **rebuttal.pdf** (Figure 1).

---

### Author Response · Authors · 2024-11-20
**Response 2**

## Comparison with CycleGAN
We compared our method with CycleGAN because we found that discrete tokens play a crucial role in the success of training generative models. Initial attempts to generate synthetic data using only continuous EEG signals, with both input and output in continuous space, resulted in trivial solutions when training GANs and transformers. We then explored a type of GAN with a similar training procedure to EEGTrans (i.e., transitioning from continuous to discrete modality), which led us to use CycleGAN. Transformers, however, proved superior in generating EEG signals as they can autoregressively fine-tune small details based on previously generated tokens, which GANs cannot do due to their one-pass generation approach. However, this advantage of transformers comes at the cost of longer generation times.

## Subject-specific Information
Regarding subject-specific information, we conducted a t-SNE analysis to visualize the differences between the representations of real and synthetic data. Specifically, we compared the representations of real data with the embeddings generated by the encoder when fed with real data. During testing, the decoder generates output starting from the SOC token, with the embedding from the encoder being the only information source available to the decoder.
We plotted data from four randomly selected subjects (subjects 1, 5, 7, and 9 from the BCI Competition IV 2a dataset) against the encoder's embeddings. In the original data, subjects tend to cluster distinctly (e.g., subject 5 in the top left corner and subject 9 in the bottom right corner). However, for the embeddings, most subjects appear spread out from the center. This indicates that, when trained across three datasets, the encoder primarily learns to retain motor imagery-relevant information for token prediction rather than capturing subject-specific details. The visualization can be found in the supplementary material **rebuttal.pdf** (Figure 2).

## Future Directions
We aim to explore the potential of generating synthetic data in an unsupervised manner, broadening the scope of our contributions. Currently, we validate that when provided with strong conditions (i.e., real data), the model can generate synthetic data with corresponding labels. Future work may involve training transformers with a mix of real data labels to generate various types of synthetic data. Our goal is not to limit our work to data augmentation but rather to explore whether transformers can generate synthetic datasets from unseen datasets, given sufficient conditions. Although not explicitly trained on target datasets, we evaluate performance using the motor imagery paradigm's classification tasks, which remain the most straightforward approach.

## Token Prediction Convergence
The next-token prediction for EEGTrans converged with a token prediction accuracy of 85% on the first testing dataset (BCI Competition IV 2a) after approximately 75 epochs of training. The visualization can be found in the supplementary material **rebuttal.pdf** (Figure 3).

## Anonymous GitHub
We apologize for the incomplete link provided earlier. You can access our code at the following link: EEGTrans: Transformer-Driven Generative Models for EEG Synthesis. (https://anonymous.4open.science/r/EEGTrans-Transformer-Driven-Generative-Models-for-EEG-Synthesis-0FD9/README.md)

---

### Meta-Review · Area_Chair_tHXa · 2024-12-23

**Metareview:**

This paper introduces a new generative model for EEG called EEGTrans. The main idea behind the work is to generate new EEG data to improve model generalization.

The reviewers agreed that the topic is important, however, multiple reviewers found the methods description unclear and the evaluations were insufficient. In particular, the reviewers pointed out that the paper was missing critical comparisons to existing baselines for both classification and generation (like EEG-GAN), as well as other forms of time-series augmentations used in the field. Additionally, there were concerns that the proposed forecasting capabilities of the model were not evaluated appropriately and that existing methods for evaluating data generation quality like PSD, STFT, activation in alpha and beta bands, or source analysis have not been carried out.

**Additional Comments On Reviewer Discussion:**

The authors provided a general response but did not provide adequate replies to each reviewer. The reviewers were all in consensus that the paper is not ready for publication.

---

### Decision · Program_Chairs · 2025-01-22

Reject